# The Gut Microbiome, Microsatellite Status and the Response to Immunotherapy in Colorectal Cancer

**DOI:** 10.3390/ijms24065767

**Published:** 2023-03-17

**Authors:** Toritseju O. Sillo, Andrew D. Beggs, Gary Middleton, Akinfemi Akingboye

**Affiliations:** 1Institute of Cancer and Genomic Sciences, College of Medical and Dental Sciences, University of Birmingham, Birmingham B15 2TT, UK; 2Institute of Immunology and Immunotherapy, College of Medical and Dental Sciences, University of Birmingham, Birmingham B15 2TT, UK; 3Dudley Group NHS Foundation Trust, Dudley DY1 2HQ, UK

**Keywords:** mismatch repair, colorectal cancer, microbiome, immunotherapy, immune environment

## Abstract

There is increasing evidence in a range of cancer types that the microbiome plays a direct role in modulating the anti-cancer immune response both at the gut level and systemically. Differences in the gut microbiota have been shown to correlate with differences in immunotherapy responses in a range of non-gastrointestinal tract cancers. DNA mismatch repair-deficient (dMMR) colorectal cancer (CRC) is radically different to DNA mismatch repair-proficient (pMMR) CRC in clinical phenotype and in its very good responses to immunotherapy. While this has usually been thought to be due to the high mutational burden in dMMR CRC, the gut microbiome is radically different in dMMR and pMMR CRC in terms of both composition and diversity. It is probable that differences in the gut microbiota contribute to the varied responses to immunotherapy in dMMR versus pMMR CRC. Targeting the microbiome offers a way to boost the response and increase the selection of patients who might benefit from this therapy. This paper reviews the available literature on the role of the microbiome in the response to immunotherapy in dMMR and pMMR CRC, explores the potential causal relationship and discusses future directions for study in this exciting and rapidly changing field.

## 1. Introduction

The role of the commensal microbiota in gut development, maintaining integrity, metabolism and immunity is critical. There are over 10^13^ commensals in the human gut [1]. The majority (99%) of species are bacterial, with Bacteroides and Firmicutes predominating, but there are also viruses, archaea and eukarya [2]. There is a bidirectional interaction between the gut microbiome and the host’s genetics [3], with added interactions from environmental factors, lifestyle and dietary changes [4]. There is a significant diversity in the microbiota amongst healthy individuals, and between healthy people and those with certain conditions. There is an extensive interaction between the gut microbiota and the gut immune environment, which modulates the response to and facilitates the tolerance of commensal bacteria and food antigens while allowing the recognition of and immune responses to harmful antigens. Dysbiosis, or an imbalance in the microbiome, can drive the development of some diseases. In particular, patients with inflammatory bowel disease are noted to have reduced faecal microbiota diversity [5]. A differential microbial gene expression is present in patients with ulcerative colitis and primary sclerosing cholangitis-associated colitis compared with healthy controls, with the dysregulation of bile acid synthesis pathways and the upregulation of Th17- and IL17-producing CD4 T cells, which are drivers of inflammation in autoimmunity [6].

The gut mucosa is composed of an epithelial layer with intestinal epithelial cells and intraepithelial lymphocytes. The lamina propria, which lies beneath this, is a connective tissue layer with lymphoid nodules, antigen-presenting cells, innate lymphoid cells, T cells and B cells. This is by far the largest component of the immune system, and the interactions at this interface are believed to extend beyond the level of the gut mucosa to the immune system as a whole [7]. Understanding the mechanisms of microbiota–immune system interactions will be helpful in identifying potential therapeutic targets for the remission of inflammatory disease and for boosting the response to immunotherapy in cancer.

The gut microbiome is hypothesised to have a crucial role in the development of colorectal cancer (CRC), with evidence that gut microorganisms modulate colorectal tumorigenesis. In animal models, specific microbes associated with colonic inflammation can drive carcinogenesis [8]. Bacteroides fragilis rapidly induces colitis and colon tumours in mice heterozygous for the APC gene, with a marked downregulation of effector T-cell responses and the upregulation of regulatory T-cell responses [8,9]. In particular, certain bacterial strains, most notably *Fusobacterium nucleatum*, *Escherichia coli*, *Bacteroides fragilis* and *Salmonella enterica*, are detected in human biopsies in gastrointestinal cancers, and they could be considered high risk factors for carcinogenesis [10]. *Fusobacterium nucleatum* has been shown to cause tumorigenesis in animal models [11]. Another microbe, *Streptococcus gallolyticus*, is also strongly associated with human CRC [12]. *S. gallolyticus* has been shown to be strongly associated with colon tumour cells and to promote tumorigenesis through the upregulation of *wnt* signalling pathways [13]. A meta-analysis of faecal metagenomes detected some global signatures specifically associated with CRC, including *Fusobacterium*, *Porphyromonas*, *Parvimonas*, *Peptostreptococcus*, *Gemella*, *Prevotella* and *Solobacterium* [14].

Gut microbiota differ significantly between patients with CRC and healthy controls [15,16]. In patients with CRC, there are also differences between mucosal and faecal samples [15,16], as well as in the tumours of patients with lower-stage versus higher-stage disease [17]. Certain gene mutations, notably *APC* and *KMT2C*, are associated with differing microbial profiles. These differences in the gut microbiome profile also extend to different molecular and genomic subtypes of CRC [18,19].

There is some evidence of alterations in the microbiome depending on the primary tumour location, with variances seen in the relative abundance of specific bacteria and bacteria-derived metabolites in colon versus rectal cancer [20,21]. Clinical data show that patients with rectal and left-sided colon cancers have superior outcomes to those with right-sided cancers [22], although these may reflect differences in treatment rather than genomic or metagenomic phenomena. The Cancer Genome Atlas data show that colon and rectal cancers are genomically very similar [23], but there are gradual molecular, genomic and phenotypic changes along the length of the large bowel, particularly with gradual increases in the frequencies of microsatellite instability, *BRAF* mutations and the CpG island methylator phenotype from the rectum to the caecum [24]. These phenotypic changes may be due to a gradual alteration in the microbiome from the proximal to the distal colon [24].

The gut bacterial flora composition also appears to have a direct impact on the anti-cancer immune response. In a mixed in vitro and in vivo CRC mouse model, exposure to certain bacterial species enriched in colonic tissues, particularly *Fusobacterium nucleatum*, *Bacteroides fragilis* and *Escherichia coli*, stimulated Th1 chemokine production, which induced T-cell tracking into the tumour [25]. This was associated with a tumour burden reduction and improved survival. The authors noted differences in the microbiota in CD3-high compared with CD3-low tumours, with *Alloprevotella, Treponema* and *Desulfovibrio* enriched in the CD3-high tumours, but *Prevotella, Bacteroides* and *Fretibacterium* were over-represented in the CD3-low tumours. The treatment of tumour-bearing mice with ampicillin and vancomycin antibiotics significantly reduced tumour-derived chemokine expression, with reductions in specific Bacteroidales and Firmicutes bacteria families and worsened outcomes. They inferred that commensal bacteria are the main chemokine inducers in CRC cells, and differences in the microbiome profile can lead to differences in prognosis [25].

These differences in the gut microbiota are relevant in predicting the responses to therapies, particularly immunotherapy, which has emerged as an effective treatment in a range of solid tumours [26,27]. There is evidence that differences in the microbiome are implicated in determining the response to immunotherapy, particularly in melanoma [7,28,29,30]. In CRC, where there are marked differences in the response to immunotherapy in mismatch repair-deficient and mismatch repair-proficient tumours [26], the finding that there are differences in the microbiome in microsatellite-stable versus microsatellite-unstable CRC [19] offers a potential explanation for this response. This paper reviews the available evidence of the role of the microbiome in shaping the immune environment and the response to immunotherapy in CRC, with particular focus on the interaction with mismatch repair status. It also reviews the roles that the rapid developments in next-generation sequencing, particularly the increasing use of shotgun whole-genome sequencing techniques, have played in expanding the field of microbiome research and in bringing it closer to clinical practice. Ongoing clinical trials of microbiome modulation in CRC are analysed, and directions for future research are recommended. The results provide a compelling argument for including an assessment of the microbiome as a molecular marker when stratifying patients for targeted therapies.

## 2. Methods

A literature search was conducted using the PubMed and MEDLINE databases and reference lists from appropriate papers. An overview of published research in the field of metagenomics and colorectal cancer immunotherapy is provided, with a focus on the interaction with mismatch repair status. The following keywords were used to perform flexible searches within these databases: ‘colo*’, ‘cancer’, ‘immunotherapy’, ‘metabolo*’, ‘metagenomic’, ‘microsatellite’, ‘microbiome’ and ‘mismatch repair’. The search was performed between November 2022 and January 2023. Only papers published in English and citable were included.

## 3. Results

### 3.1. Mismatch Repair-Deficient and -Proficient Colorectal Cancer

About 15% of colorectal cancers demonstrate microsatellite instability (MSI) secondary to mutations in DNA mismatch repair (MMR) genes (MLH1, MSH2, MSH6 and PMS2) [31,32]. Of these, about three-quarters are sporadic, with no underlying germline mutations, and they arise as a consequence of somatic mutations in the mismatch repair genes or the epigenetic silencing of the MMR gene MLH-1 by the hypermethylation of its promoter region [33]. Lynch syndrome is found in 2–3% of cases. It is caused by an inactivating germline mutation of one or more of the MMR genes, with a second hit from a sporadic mutation, a loss of heterozygosity or the epigenetic silencing of a second MMR gene [34]. These mismatch repair-deficient (dMMR) tumours have notable clinical features, including a higher preponderance in right-sided colon tumours, a higher prevalence in older patients, an earlier disease stage and a better clinical prognosis [31].

MSI-high tumours are characterised by a high mutational burden and large numbers of tumour-specific antigens (neoantigens), which are strongly immunogenic [31,32]. In Phase II clinical trials, patients with MSI-high (dMMR) CRC have been shown to have significant pathological and clinical responses to immunotherapy with immune checkpoint blockade agents, such as anti-programmed cell death 1 (PD-1)/anti-programmed cell death ligand 1 (PD-L1) treatment [26,35] (Table 1). A recent Phase II study of neoadjuvant PD-1 blockade (dostarlumab) in locally advanced rectal cancer showed 100% complete clinical responses in all 12 patients, with no one having surgery or a relapse, with a median follow-up duration of 12 months [36]. The Nivolumab, Ipilimumab and COX2-inhibition in Early Stage Colon Cancer (NICHE) trial assessed neoadjuvant immunotherapy in both dMMR and pMMR non-metastatic CRC [37]. In total, 19 of 20 patients with dMMR tumours had major pathological responses (MPRs) to treatment, with a pathological complete response (pCR) in 12 of 20 patients. In the NICHE-2 study, the results of which have recently been discussed [38], of 112 enrolled patients with dMMR CRC, 95% had MPRs, and 67% had pCR. There was a marked difference in response in those with sporadic dMMR versus those with Lynch syndrome, with patients with Lynch syndrome having a higher pCR rate than those with sporadic tumours (78% versus 58%, *p* = 0.056). In contrast to their response to immunotherapy, dMMR tumours respond poorly to neoadjuvant treatment with standard chemotherapy regimens [39].

For CRC that develops due to a chromosomal instability (MMR-proficient (pMMR) or microsatellite-stable (MSS) CRC), tumour mutational burden and neoantigen numbers are lower [40,41]. In this group of patients, initial clinical trials showed a minimal objective response to immunotherapy [26,35]. However, two recent studies of dual neoadjuvant immunotherapy with immune checkpoint blockade in MSS CRC have shown some promise. In the NICHE trial, 4 of 15 patients (27%) with pMMR tumours had detectable pathological responses, with 3 of these being MPRs [37]. These patients’ tumours had higher levels of CD8+PD-1+ T-cell infiltration but not an increased tumour mutational burden (TMB) when compared with those of non-responders. The Canadian Cancer Trials Group CO.26 study assessed the effect of combined immune checkpoint inhibition with anti-cytotoxic T-lymphocyte-associated protein (CTLA)-4 and anti-PD-L1 blockade compared with that of the best supportive care alone in patients with advanced CRC [42]. The median overall survival (OS) was 6.6 months in those who had dual checkpoint blockade and 4.1 months in those who had the best supportive care (HR 0.72; 90% CI, 0.54–0.97; *p* = 0.07). The OS was significantly improved with dual checkpoint blockade in MSS CRC, particularly in those with TMB of 28 variants per megabase or more. These studies suggest that other factors beyond TMB can predict responses to immunotherapy in pMMR CRC (Table 1).

**Table 1 ijms-24-05767-t001:** Clinical trials of immunotherapy in CRC (adapted and expanded from Sillo et al. [43]).

Phase	Reference (Trial Name)	Regimen	Subgroups	Outcomes	Follow-Up Duration
Phase II	Le et al., 2015 [26]	PD-1 inhibitor (pembrolizumab)	dMMR/MSI-high vs. MSS CRC	Immune-related ORR PFS	20 weeks
Phase II	Overman et al., 2018 [35] (CheckMate 142)	PD-1 inhibitor (nivolumab) +/− CTLA-4 inhibitor (ipilimumab)	Metastatic pre-treated dMMR/MSI-high CRC	Immune-related ORR PFS OS	12 months
Phase II	Mettu et al., 2022 [44] (BACCI)	Capecitabine/bevacizumab +/− PD-L1 inhibitor (atezolizumab)	Metastatic CRC	PFS OS	20.9 months
Phase III	Eng et al., 2019 [45] (COTEZO Imblaze370)	Cobimetinib + PD-L1 inhibitor (atezolizumab) vs. atezolizumab vs. regorafenib	Heavily pre-treated locally advanced or metastatic CRC (>95% MSS)	OS PFS	3 years
Phase III	Diaz et al., 2022 [46] (KEYNOTE-177)	PD-1 inhibitor (pembrolizumab) vs. standard chemotherapy	dMMR/MSI-high Stage 4 CRC	PFS OS	44.5 months
Phase III	Kim et al. [47] (POLE-M)	Standard 5-FU-based adjuvant chemotherapy +/−sequential PD-L1 inhibitor (avelumab)	Resected stage 3 dMMR/MSI-high or POLE-mutant colon cancer	DFS	16.3 months
Phase III	Sinicrope et al., 2017 [48] (ATOMIC, Alliance A021502)	Combined chemotherapy +/− PD-L1 inhibitor (atezolizumab) as monotherapy for additional 6 months	Resected stage 3 dMMR/MSI-high colon carcinomas	DFS OS Adverse events	5 years
Phase I	Tabernero et al., 2017 [49]	CEA-TCB antibody +/− PD-L1 inhibitor (atezolizumab)	Heavily pre-treated metastatic CRC (majority MSS)	Adverse events Anti-tumour activity (RECIST v1.1 criteria [50]) PFS	40 months
Phase I (exploratory)	Chalabi et al., 2020 [37] (NICHE)	Combined PD-1 inhibitor (nivolumab), CTLA-4 inhibitor (ipilimumab) +/− COX2 inhibition, then surgery	dMMR and pMMR CRC, neoadjuvant, stage 1 to 3 disease only	Adverse events Immune-activating capacity RFS	3–5 years (ongoing)
Phase II	Antoniotti et al., 2020 [51] (AtezoTRIBE)	Combined 5-FU-based chemotherapy + bevacizumab + PD-L1 inhibitor (atezolizumab) vs. combination treatment	Unresected and previously untreated metastatic CRC, irrespective of MMR status	PFS Overall toxicity rate ORR	24 months (ongoing)
Phase II	Chen et al., 2020 [42] (Canadian Cancer Trials Group CO.26)	Combined PD-L1 (durvalumab) and CTLA-4 inhibitor (tremelimumab) with based supportive care vs. best supportive care	Pre-treated metastatic dMMR and pMMR CRC	OS PFS ORR	15.2 months
Phase II	Cercek et al., 2022 [36]	Neoadjuvant PD-1 (dostarlimab) followed by chemoradiotherapy and surgery	dMMR rectal cancer	Complete clinical response	6 to 25 months
Phase I	Chalabi et al., 2022 [38] (NICHE-2)	Combined PD-1 inhibitor (nivolumab), CTLA-4 inhibitor (ipilimumab), then surgery	dMMR and CRC, neoadjuvant, stage 1 to 3 disease only	Safety DFS	Ongoing

CRC = colorectal cancer. CTLA-4 = cytotoxic T-lymphocyte-associated protein 4. DFS = disease-free survival. dMMR = deficient mismatch repair. PD = programmed cell death 1. PD-L1 = programmed cell death ligand 1. PFS = progression-free survival. pMMR = proficient mismatch repair. ORR = objective response rate. OS = overall survival. RECIST = Response Evaluation Criteria in Solid Tumours. RFS = recurrence-free survival.

### 3.2. Distinct Microbiomes in Mismatch Repair-Deficient and -Proficient Colorectal Cancer

The differences in the immunogenomic features and responses to therapies observed in dMMR and pMMR CRC have been thought to be caused by somatic factors, such as tumour mutational burden [27]. Yarchoan et al. [41] correlated the response rates to anti-PD-1 or anti-PD-L1 blockade with non-synonymous tumour mutational burden in a range of tumour types. They found a strong linear association with the median number of coding somatic mutations per megabase pair and the objective response rate (ORR) as defined by RECIST criteria (Response Evaluation Criteria in Solid Tumours) to immunotherapy. Tumour types with high mutational loads, for example, melanoma and dMMR CRC, had high response rates (>40%), and those with low mutational loads, such as pancreatic cancer and sarcoma, had much lower response rates. pMMR CRC was a curious outlier. Although on average the median mutational load was lower than that in dMMR CRC, there was a marked paucity of the response to anti-PD-1 inhibition. It is highly likely that other factors attenuate this response.

A few studies have specifically analysed and compared the microbiome in dMMR and pMMR CRC and provided evidence that there are distinct phenotypes. In a study of 83 patients who underwent total or partial colectomy for CRC, MMR status was a strong predictor of microbial community variance [19]. Common CRC-associated microbial signatures, particularly *Fusobacterium* spp. and *Bacteroides fragilis*, were enriched in dMMR compared with pMMR tumours. They also used a combination of metabolomics and metabolic modelling to demonstrate that dMMR CRC had a greater hydrogen sulphide production, while pMMR CRC had a greater metabolic suppression of *Bacteroides fragilis*. MMR status was also revealed to be more strongly correlated with microbial community differences than other measures, including sample location, sample type, age, sex and body mass index. This was supported by data from Jin et al. [52], who, in a study of 230 patients, found a greater richness (alpha diversity) in dMMR compared with pMMR CRC, as well as differences in the microbial composition (beta diversity), with an enrichment of *Fusobacterium, Akkermansia, Bifidobacterium, Faecalibacterium, Streptococcus* and *Prevotella* at the genus level in dMMR tumours. A functional analysis of these intestinal flora using Kyoto Encyclopaedia of Genes and Genomes (KEGG) pathways revealed clearly different metabolic pathways in the dMMR and pMMR samples.

CRC is often grouped into consensus molecular subtypes (CMSs) based on immunogenomic profiles [53]. The CMS1 subtype, which is characterised by a highly inflamed tumour immune microenvironment, is highly associated with dMMR CRC. Purcell et al. [18] used RNA-sequencing-derived gene expression profiling to classify 34 tumours into CMS subtypes, and they performed a 16S ribosomal RNA (rRNA) amplicon analysis to detect specific metagenomic signatures in the CMS1 group. At the level of bacterial phyla, there was a strong preponderance of Fusobacteria and Bacteroidetes and decreased levels of Firmicutes and Proteobacteria, in keeping with the findings in dMMR CRC in other studies [19,52] (Table 2). Two studies specifically analysing the quantity of *Fusobacterium nucleatum* DNA in CRC samples consistently found raised levels in dMMR CRC compared with those in pMMR CRC ([54,55], Table 2). Tahara et al. [54] found that *Fusobacterium* was higher in tumours than in adjacent colonic tissue, but samples with very high levels had a specific phenotype, which corresponded to the dMMR/MSI-high subgroup. Mima et al. [55] similarly found a significant association between microsatellite instability and high levels of *Fusobacterium* DNA, independent of methylation status and BRAF mutation status, in a group of over 1000 colorectal tumour samples.

A study assessing polyp formation in a dMMR mouse model showed that altering the gut microbiota via antibiotic administration significantly reduced early-stage colonic polyp numbers in *MSH2*-deficient mice, suggesting that these microbiota act at an early stage in the development of dMMR CRC [56]. Further metabolomics analyses showed that certain luminal metabolites, particularly butyrate, were reduced in these treated mice. They propose that gut microbiota-produced metabolites, such as butyrate, drive the proliferation of aberrant epithelial cells through the *wnt*/β-catenin pathway in dMMR CRC [57]. It is thus possible to infer that certain microbiota, particular *Fusobacterium*, are tumorigenic in the context of dMMR but can also exert beneficial effects in enhancing the response to immune therapies.

An important histological subset of CRC is mucinous adenocarcinoma. It is present in 10–15% of patients [58]. Mucinous tumours show more than 50% extracellular mucin in histological assessments. They have a predilection for the proximal colon [59], and they are strongly associated with inflammatory bowel disease and microsatellite instability [60]. They are also associated with higher rates of *RAS/RAF/MAPK* and *AKT/PI3K* pathway mutations [61]. Mucinous tumours are usually associated with a poorer prognosis, although dMMR-associated mucinous tumours have been seen to have a better prognosis [62]. The molecular and genomic drivers of mucinous tumours are unclear, and it is possible that the microbiome has a role in aberrant mucin production in these tumours. In the healthy colon, mucin is produced by colonic epithelial cells, and it forms an apical barrier, shielding the epithelium from physical and chemical injury caused by food and microbes. In mucinous tumour cells, excessive mucin completely surrounds the cell surface, acting as a physical barrier that assists with evading anti-tumour immune mechanisms and providing an adhesive conduit for tumour cell tracking to endothelia and from there to distant structures [60].

The formation of bacterial biofilms, particularly those formed by *E. coli* and *B. fragilis*, may enhance mucin production [60]. Mucinous tumours overexpress MUC2, a secreted gel-forming mucin that is encoded in the *MUC2, MUC5AC, MUC5B* and *MUC6* genes on chromosome 11 [63]. MUC5AC is a paralog of MUC2 that is expressed by gastric and airway epithelia. The ectopic expression of MUC5AC is observed in precancerous and cancerous colonic mucosae. MUC6 is a gastric mucin that is ectopically expressed in colorectal cancer. Bacteria may increase MUC2 expression and upregulate pro-inflammatory cytokines that drive mucin production, as seen in human colonic adenocarcinoma cells incubated with *F. nucleatum* [64]. It is still not known if there are distinct microbial signatures in mucinous CRC, and this represents a potential area for further study.

### 3.3. Gut Microbiota and the Anti-Tumour Immune Response

The composition of the gut microbiome is implicated in immune regulation and in inter- and intra-tumour heterogeneity in tumour immunity. Several studies have shown clear associations between microbiome diversity and the response to immunotherapy, in both pre-clinical (animal models) and clinical studies [7,29,30,65,66,67]. An alteration in the gut microbiome by mechanisms such as the administration of systemic antibiotic therapy or faecal microbiota transplantation (FMT) leads to notable differences in the response (Table 3).

Two pre-clinical models in 2015 showed that anti CTLA-4 and anti PD-1 blockade were effective in mice with particular bacterial strains [65,66]. In the first, Sivan et al. demonstrated that CTLA-4 blockade was only effective in sarcomas induced in mice that carried specific *Bacteroides* species in their intestinal flora [65]. This effect was lost in antibiotic-treated mice. A cause-and-effect relationship was determined by observing that recolonising antibiotic-treated and germ-free mice with these specific species recovered the anti-cancer response, with a reduction in tumour growth and the infiltration of antigen-specific T cells into the tumour. This response was of a similar degree to that with anti-PD-L1 immunotherapy. Similarly, in a melanoma model, anti-PD-1 therapy was shown to be effective in *Bifidobacterium*-treated mice [66], with a specific *Bifidobacterium* operational taxonomic unit (OTU_681370) having the strongest associations with anti-tumour T-cell responses. The results were corroborated by a study in mice inoculated with an MSS-type colon carcinoma cell line (CT26) [67]. The injection of a combination of broad-spectrum antibiotics (ampicillin, streptomycin and colistin), which eradicated the gut microbiome, compromised the effect of the anti-mouse PD-1 inhibitor and promoted tumour growth. A faecal microbiome analysis prior to treatment revealed that the gut microbiome diversity was associated with the response to anti-PD-1 immunotherapy in these mice, with *Prevotella* spp. and *Akkermansia muciniphila* being related to the better efficacy of immunotherapy and *Bacteroides* spp. being related to the poor efficacy of immunotherapy. Further evidence from Zhuo et al. [68] showed that a combination of Lactobacillus acidophilus lysates and anti-CTLA-4 blockade enhanced the anti-tumour response in a colitis-induced CRC mouse model.

These results are strongly supported by evidence from clinical studies, predominantly in melanoma and non-colorectal cancers. Gopalakrishnan et al. showed that gut microbiome diversity was positively correlated with ORRs to anti-PD-1 therapy in patients with melanoma [7]. Stool sample diversity was positively correlated with improved progression-free survival (PFS). In particular, PFS was higher in patients with an abundance of *Faecalibacterium* spp. but lower in those with a high preponderance of the Bacteroidales order. Matson et al. [30] used 16s rRNA gene amplicon sequencing to show that stool samples from patients with metastatic melanoma who responded to immunotherapy had an abundance of certain bacterial species, notably *Bifidobacterium longum, Collinsella aerofaciens* and *Enterococcus faecium*, while non-responders had an abundance of *Ruminococcus obeum* and *Roseburia intestinalis*. Flow cytometry and cytokine assays from patients showed that those with a high abundance of favourable microbes (including Clostridiales, Ruminococcaceae and Faecalibacterium) had higher densities of effector T cells (CD4+ and CD8+) in the systemic circulation, while those with higher frequencies of Bacteroidales had more regulatory T cells and myeloid-derived suppressor cells. A germ-free mouse tumour model also demonstrated similar responses to FMT from responders [30].

In patients with a range of epithelial cancers, predominantly non-small-cell lung cancer and renal cell carcinoma, Routy et al. found that an abnormal gut microbiome composition could be responsible for non-response to anti-PD-1 immunotherapy [29]. The administration of systemic antibiotic treatment just prior to commencing immunotherapy led to a worsened PFS and a reduced overall survival compared to those in a comparable non-treated group. This was postulated to be due to the alteration of the gut microbiome by antibiotic therapy. In a selection of 100 patients from this cohort, quantitative metagenomics was used and compared to a reference catalogue of the human microbiome genome [69]. Responders to immunotherapy had microbe profiles that differed to those of non-responders, with an abundance of *Akkermansia municiphilia, Enterococcus hirae* and *Alistipes indistinctus* in the responders. Furthermore, FMT from the responders into germ-free or antibiotic-treated mouse tumour models led to significant anti-tumour responses, with the upregulation of dendritic cell and effector T-cell responses. This did not occur with FMT from the non-responders (Table 3).

A potential explanation for these results is that beneficial microbiota (such as *Bifidobacterium*, *Bacteroides* and *Akkermansia* spp.) interact with and co-operate with immune checkpoint inhibitors at the gut epithelial surface level, increasing antigen-presenting cell (particularly dendritic cell) activation, boosting anti-tumour effector T-cell responses and suppressing regulatory T-cell responses ([70], Figure 1). This has both local activity at the primary tumour site and systemic effects due to the trafficking of these effector cells through lymphatic channels to distant sites, leading to more favourable clinical outcomes.

### 3.4. Therapeutic Manipulation of the Microbiome

The observed differences in the microbiome and metabolic landscapes in dMMR and pMMR CRC, with a potential causal association between the microbiome and the colorectal tumour environment, raise the attractive possibility that responses to immune checkpoint blockade can be boosted by altering the microbiome diversity in patient populations.

FMT is an attractive option for the manipulation of the microbiome, as it is cost-effective in the treatment of *C. difficile* colitis, with an acceptable safety profile [71,72]. It is also effective in inflammatory bowel disease-associated colitis [73]. Most studies of FMT in cancer immunotherapy have been performed in melanoma [74]. These have provided encouraging results, with the transfer of microbiota from responders to non-responders inducing good clinical responses [75,76]. In one study, objective responses were observed in 3 of 10 previously non-responding patients with metastatic melanoma [75], with changes to a more favourable immune profile seen. In another, good clinical responses were seen in 6 of 15 patients with refractory metastatic melanoma following FMT from responders [76]. These suggest that a simple mucosal transplantation of favourable microbes is sufficient to induce anti-tumour immune responses, which are enhanced by immune checkpoint blockade [28].

Thus far, there are no approved probiotic or FMT regimes in clinical use in the treatment of CRC. However, there are currently two early Phase II trials of FMT in CRC, which are in the recruitment phase. The M.D. Anderson Cancer Centre Phase II study aims to assess the effect of the re-introduction of anti-PD-1 therapy in non-responders with dMMR CRC after FMT from responders to non-responders [77]. The primary outcome will be the ORR by RECIST criteria for up to 3 years after treatment. The second study, based at the Cancer Institute and Hospital, Chinese Academy of Medical Sciences, aims to recruit 30 patients and will assess the efficacy of a combination of FMT, immune checkpoint inhibition and a tyrosine kinase inhibitor in patients with refractory advanced CRC [78]. The results of these studies are anticipated, as they could transform and widen the therapeutic options for patients for whom immunotherapy is either currently not licensed or who have refractory disease. They will also provide information on the effects of FMT on the immune profile and response to immunotherapy in dMMR and pMMR CRC.

### 3.5. Next-Generation Sequencing and Advances in Metagenomics

Breakthroughs in the understanding of the role of the microbiome in the innate response to cancer and to targeted therapies have been driven by major advances in genomic sequencing over the past few decades. Metagenomic studies have traditionally used 16S rRNA sequencing to identify the presence of bacteria and archaea [79]. These studies rely on the ubiquity and essential function of the 16S rRNA gene. The sequence identity is used to determine bacterial species similarity, with an operational taxonomic unit (OTU) defined as organisms displaying 97–98% identify in the 16S rRNA gene. 16s rRNA sequencing is cost-effective, and data analyses can be readily performed using established pipelines [80]. However, there are some disadvantages. There may be a significant microdiversity, even amongst organisms with clusters of sequences with a very high sequence identity [79,81]. Some bacteria also have several copies of the rRNA gene, which can lead to biases in the results [79]. Mechanical errors introduced during sample handling and fixation and inherent biases associated with the polymerase chain reaction required for amplifying the marker genes in 16S rRNA sequencing reduce the reliability of results obtained [82]. The reference gene catalogues on which the analyses are based are often derived from single-cohort samples, which limits the coverage of the global microbiome diversity [69]. It also does not provide information about non-bacterial microbes (viruses, archaea and protozoans).

Next-generation sequencing techniques are increasingly used due to their many advantages over 16s rRNA techniques. Shotgun whole-genome sequencing (WGS) relies on shotgun sequencing with random primers to sequence overlapping regions of a genome [80]. It generates a greater accuracy of taxonomic definitions at the species level, increased genomic diversity and a greater detection of bacteria and non-bacterial species, and it can provide evidence of gene functional variation in species [83,84]. Non-bacterial genomes are also identified, which raises the possibility of investigating the roles of these organisms within the gut and systemic microbiome. Viral metagenomics is an evolving field. A recent analysis by the Pan-Cancer Analysis of Whole Genomes (PCAWG) Consortium explored associations between tumour-associated viruses (notably Epstein–Barr virus, human papilloma virus (HPV) and hepatitis B virus) and many cancer types [85]. They found exclusivity between the presence of HPV genomes and driver mutations in head and neck cancers. However, they found no significant associations between these viruses and colon/rectum tumours. Overall, the published evidence on the link between viruses and CRC remains limited and uncertain [86]. Further viral metagenomic studies will help to define any potential causative links.

There are challenges with shotgun WGS. As all DNA in a sample is indiscriminately sampled, it requires a greater sequencing depth to determine unique taxonomic identifiers [87], leading to increased costs, although these are continuing to fall [88]. Metagenomic WGS techniques also rely on reference databases, which makes the identification of novel microbes more challenging without computationally intensive pipelines and increases the susceptibility to false-positive results [84,89].

## 4. Discussion

There have been recent developments in the understanding of the role of the microbiome in the response to cancer and anti-cancer therapies. In CRC, this is of significant interest given the proximity of the primary tumour to the gut microbiome. Advances in sequencing techniques, which increase the diversity of genomes discovered and reduce bias in the results obtained, continue to enable rapid developments in our understanding of the roles of bacterial and non-bacterial genomes within the gut and systemic microbiome, and their associations with carcinogenesis and clinical outcomes.

These studies have uncovered the roles of microbial diversity in tumour evolution, tumour propagation and the immune response to cancer. The finding that gut microbiota directly impact the response to immunotherapy in a range of cancer types and that this response can be modified by changes to the microbiome offers an attractive method for improving clinical responses. This has led to interest in modifying the microbiome to improve responses to therapies and clinical outcomes [74].

However, there are some challenges in interpreting the observed results. Heterogeneity in microbiota composition in tumours and the adjacent colon, in faecal versus mucosal samples and in primary tumour location add complexity to comparisons of results across different studies and could limit their generalisability [15]. The microbiome is directly influenced by environmental events, including local and systemic antibiotic therapy, the use of probiotics and dietary intake, which makes the standardisation of any study or intervention challenging [90]. There are some apparently contradictory effects of the gut microbiota in CRC. *Fusobacterium nucleatum* and *Bacteroides* in particular appear to both drive colorectal tumorigenesis and be associated with improved responses to immunotherapy and better clinical outcomes. This implies that, in the presence of certain genomic markers, such as mismatch repair deficiency, some microbiome and metabolomic profiles can induce tumorigenesis and tumour propagation whilst offering protective benefits in other states. Elucidating these mechanisms is an area of key importance.

## 5. Conclusions

While it is possible to establish clear associations between specific microbiome profiles and carcinogenesis, prognosis and response to therapies, the determination of clear causative relationships is significantly more challenging. Our understanding of the interaction between the microbiome and the genome in cancer is in its early stages. Future research will harness the potential of advances in genomics and bioinformatics techniques to clearly establish causative associations between microbiome diversity and tumour evolution. The role of non-bacterial (particularly viral) genomes in cancer is an area that is ripe for future exploration. Finally, a key direction of future travel is the manipulation of the microbiome in cancer. Evidence for the efficacy of combined FMT and immunotherapy, predominantly in melanoma, can also translate to beneficial effects in patients with CRC. The early-phase trials of FMT in dMMR and refractory CRC offer hope for therapeutic potential for patients, and the results are closely anticipated. Future studies should aim to delineate the key interactions between the microbiome, cancer genomes and host genomes to aid the development of mechanisms for the manipulation of the microbiome for therapeutic benefit.

## Figures and Tables

**Figure 1 ijms-24-05767-f001:**
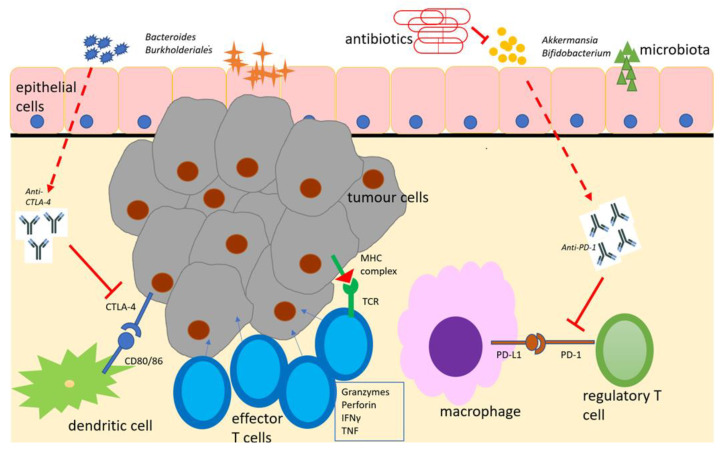
Schematic representation of the microbiome with immunotherapeutic agents (anti-PD-1 and anti-CTLA-4), antigen-presenting cells and effector cells in the tumour immune microenvironment. Adapted from [70]. Beneficial microbiota boost the effects of immune checkpoint inhibitors, which, in turn, suppress regulatory T-cell responses and enhance effector T-cell responses to encourage tumour cell lysis. Antimicrobial therapy suppresses beneficial microbiota and diminishes this response. CTLA-4 = cytotoxic T-lymphocyte antigen-4. IFNγ = interferon gamma. MHC = major histocompatibility complex. PD-1 = programmed cell death -1. PD-L1 = programmed cell death ligand 1. TCR = T-cell receptor.

**Table 2 ijms-24-05767-t002:** Microbiome and metabolomic signature differences in dMMR and pMMR CRC.

Authors	N	Samples	Microbiota Enriched in dMMR CRC	Microbiota Enriched in pMMR CRC	Metabolomic Signatures in dMMR CRC
Tahara et al., 2014 [54]	149	Paired tumour and control (adjacent) colon tissue	*Fusobacterium nucleatum*	Not assessed	Not assessed
Mima et al., 2016 [55]	1069	Tumour tissue	*Fusobacterium nucleatum*	Not assessed	Not assessed
Purcell et al., 2017 [18]	34	Tumour tissue	*Fusobacterium* *Akkermansia* *Bifidobacterium Faecalibacterium* *Streptococcus* *Prevotella*	*Serratia* *Cupriavidus* *Sphingobium*	Not assessed
Hale et al., 2018 [19]	83	Paired tumour and control (adjacent) colon tissue	*Fusobacterium* spp. *Bacteroides fragilis*		Hydrogen sulphide
Jin et al., 2022 [52]	230	Paired tumour and control (adjacent) colon tissue	*Fusobacterium* *Akkermansia* *Bifidobacterium Faecalibacterium Streptococcus Prevotella*	Proteobacteria *Serratia* *Cupriavidus Sphingobium*	Glycan biosynthesis and metabolic pathways Nucleotide metabolic pathways Cell growth and death, genetic replication and repair

CRC = colorectal cancer. dMMR = deficient mismatch repair. N = patient number. pMMR = proficient mismatch repair.

**Table 3 ijms-24-05767-t003:** Studies investigating the gut microbiome and immunotherapy response.

Authors	Model	Cancer Type	N	Regimen	Microbiome Alteration	Faecal Microbiome in Responders	Microbiome in Non-Responders
Sivan et al., 2015 [65]	Pre-clinical (mouse)	Melanoma	n/a	Anti-PD-1	FMT	*Bifidobacterium*	
Vetizou et al., 2015 [66]	Pre-clinical (mouse) and clinical	Sarcoma/melanoma	25	Anti-CTLA-4	Broad-spectrum antibiotics FMT	*Bacteroides* *Burkholderiales*	
Routy et al., 2018 [29]	Clinical	Epithelial cancers	100	Anti-PD-1/PD-L1	Prior systemic antibiotics FMT	*Akkermansia municiphilia Enterococcus hirae* *Alistipes indistinctus*	*Parabacteroides* spp. Clostridiales *Corynebacterium*
Gopalakrishnan et al., 2018 [7]	Clinical	Melanoma	112	Anti-PD-1	Nil	*Faecalibacterium* spp.	Bacteroidales
Matson et al., 2018 [30]	Pre-clinical (mouse) and clinical	Melanoma	42	Anti-PD-1	FMT	*Bifidobacterium longum* *Collinsella aerofaciens Enterococcus faecium*	*Ruminococcus obeum* *Roseburia intestinalis*
Zhuo et al., 2019 [68]	Pre-clinical (mouse)	CRC	n/a	Anti-CTLA-4	Lactobacillus acidophilus	*Proteobacteria* *Firmicutes*	*Bacteroides*
Xu et al., 2020 [67]	Pre-clinical (mouse)	MSS CRC	8	Anti-PD-1	Systemic antibiotics	*Akkermansia municiphilia* *Prevotella* spp.	*Bacteroides* spp. *Bacteroides_sp._CAG927*

CRC = colorectal cancer. CTLA-4 = cytotoxic T-lymphocyte-associated protein 4. FMT = faecal microbiota transplantation. MSS = microsatellite stable. N = patient number. PD = programmed cell death 1. PD-L1 = programmed cell death ligand 1.

## Data Availability

Not applicable.

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
