# Peer review of "The Gut Microbiome, Microsatellite Status and the Response to Immunotherapy in Colorectal Cancer"

_ijms, 2023, doi:10.3390/ijms24065767_

Round 1
Reviewer 1 Report
Good review article
Author Response
Thank you for your response to our paper. We appreciate your assessment and kind comments.
Reviewer 2 Report
The paper, The gut microbiome, microsatellite status and the response to immunotherapy in colorectal cancer, is an excellently written review of the topic. My only reservation is about the paragraph on lines 312- 327. Is there any way to reconcile the bacteria groups indicated to be beneficial or not within the studies listed? This might involve a deeper dive by the authors into the data presented in the papers.
Author Response
Thank you for review. With regard to the bacterial groups - the studies were extremely heterogeneous due to the use of both pre-clinical and clinical models and also various epithelial cancers, melanomas and colorectal cancers. Of the few papers we found where the response to immunotherapy was studied in colorectal cancer, Akkermansia species appear to be beneficial and Bacteroides species appear to be non-beneficial and we have commented on this in the body of the text in the discussion in 466-478.
Reviewer 3 Report
The Review article authored by Toritseju O Sillo et al aims to provide an overview of the relationship between gut microbiota, microsatellite status and the response of colorectal cancer to immunotherapy.
After an introduction concerning commensal microbiota, the description of the method used to document this review, the Authors addressed successively the different subtypes of colorectal cancers, the ongoing clinical trials of immunotherapy, the microbiome and metabolome signatures in colorectal cancer with microsatellite instability vs chromosome instability, and the impact of the microbiome on the anti-tumor immune response. Then, the manuscript describes the challenge to characterize microbiota composition by next generation sequencing.
The article is concise and well-articulated, the topic is of importance, thus this review would make a suitable contribution for IJMS.
Nevertheless, the manuscript raises the following concerns
The adjacent mucosa of cancer cannot be considered as “normal”, the term “control” is more accurate.
Line 48 Peyer's patches (aggregates of lymphoid nodules) are present in small intestine. In the colon, there are lymphoid nodules
Line 90 To sustain their hypothesis, the Authors might provide a reference concerning changes in microbiome composition along the length of the colon.
Method section. The Authors should indicate the date of the end of literature survey
Result section. Please, relabel “Table 3” (lines 234, 236 and 278) to “Table 2” and “ Table 2” (lines 287 and 348) to “Table 3”
What about CD47-based immunotherapy ?
Line 270-276 “Mucinous tumours overexpress MUC2, … which is encoded in the MUC2, MUC5AC, MUC5B and MUC6 genes” ? Please, rephrase. MUC5AC is a paralog of MUC2 that is expressed by gastric and airway epithelia. The ectopic expression of MUC5AC is observed in precancerous and cancerous colonic mucosa. MUC6 is a gastric mucin that is ectopically expressed in colorectal cancer. Line 272 Please change “induce” to “increase”
Line 287, Please, change “faecal mucosal transplantation” to “faecal microbiota transplantation”
The Authors may mention bacteriophages. Accordingly, bacteriophages can modulate microbiota composition but also be used for immunotherapy
Author Response
Thank you for taking the time to read our review and for your excellent comments.
3a. We have edited the comments to reflect that these are adjacent/control.
3b. Line 48 – we have edited to say ‘lymphoid nodules’.
3c. Line 90 – The reference (24) has been added as this was a hypothesis from those authors.
3d. Methods – the dates for the literature search have been added to line 129-130.
3e. The Table numbers have been changed.
3f. CD47-based immunotherapy is not licensed for use in colorectal cancer and is commonly used in haematological malignancy. (Yang et al 2023. The landscape overview of CD47-based immunotherapy for hematological malignancies. Biomarker Research). We do not feel that discussion of this is relevant to colorectal malignancy at this time.
3g. The changes requested in paragraph 270 – 276 have been made. Thank you.
3h. Changed to faecal microbiota transplantation. Thank you.
3i. We did not specifically analyse bacteriophages in this review, although we are aware that this is a novel and evolving field. The papers we have encountered have assessed its role in infections and in inducing immunity, but we found no papers analysing the role of phages in pre-clinical or clinical models. This appears to be a rapidly developing field, and warrants a separate review article.
Reviewer 4 Report
The article describes the role of the microbiome in the response to immunotherapy in dMMR and pMMR CRC. It further explores the potential causal relationship and discusses future directions for study in this exciting and rapidly changing field. The authors used PubMed and MEDLINE databases and 124 reference lists from appropriate papers to compile this article. The article is supported by two tables and one illustration with relevant information for the article. The article is well-written and easy to understand.
The figure needs to enhance as some part is not readable.
Author Response
Thank you for your review and comments. We have enhanced the figure to improve its readability.